Transiently elevated expression of different forms of brain-derived neurotrophic factor in the neonatal medial prefrontal cortex affected anxiety and depressive-like behaviors in adolescence

Lanshakov Dmitriy lanshakov@bionet.nsc.ru 1 2
Shaburova Elizaveta 1
Sukhareva Ekaterina 1
Bulygina Veta 3
Drozd Uliana 1
Larionova Irina 4
Gerashchenko Tatiana 4
Shnaider Tatiana 5
Denisov Evgeny V. 4
Kalinina Tatyana 2 3
1 Postgenomics Neurobiology Laboratory, Institute of Cytology and Genetics , Novosibirsk , Russia
2 Novosibirsk State University , Novosibirsk , Russia
3 Functional Neurogenomics Laboratory, Institute of Cytology and Genetics , Novosibirsk , Russia
4 Laboratory of Cancer Progression Biology, Tomsk National Research Medical Center , Tomsk , Russia
5 Sector of Genomic Mechanisms of Ontogenesis, Institute of Cytology and Genetics , Novosibirsk , Russia
Badenhorst Shaw
Electronic publication date: 2024 Nov 27
Publication date: 2024
Volume: 12
Electronic Location ID: e18465
Received 2024 Jun 17; Accepted 2024 Oct 15
Copyright: ©2024 Lanshakov et al.
Copyright year: 2024
Copyright holder: Lanshakov et al.
License: This is an open access article distributed under the terms of the Creative Commons Attribution License, which permits unrestricted use, distribution, reproduction and adaptation in any medium and for any purpose provided that it is properly attributed. For attribution, the original author(s), title, publication source (PeerJ) and either DOI or URL of the article must be cited.
License URL: https://creativecommons.org/licenses/by/4.0/

Keywords: Prefrontal cortex, Neonatal period, Juvenile period, RNAseq, Anxiety, Depression

Funding: Russian Science Foundation No. 24-25-00154 Ministry of Education FWNR-2022-0002 Lentivirus production and RNAseq was funded through the Russian Science Foundation grant No. 24-25-00154. Animal care was funded through a Ministry of Education grant No. FWNR-2022-0002. The funders had no role in study design, data collection and analysis, decision to publish, or preparation of the manuscript.

==============================
Brain-derived neurotrophic factor (BDNF) is a secreted molecule that plays an important role in the survival and growth of nerve cells. BDNF undergoes complex post-translational processing with cellular proteases. Pro- and mature BDNF forms bind to different receptor types in the brain. BDNF is prominent in the neonatal cerebral cortex. The neonatal period is critical for the proper development of the brain and nervous system. Disruptions in these critical periods can have long-lasting effects on behavior and mental health. Individuals who experience adverse effects in the neonatal period have demonstrated a predisposition to depression and other neurobehavioral disorders. In this work we studied the influence of transient expression (P3–P8) elevation of pro-, mature and mutant forms of BDNF that could not be processed with cellular convertases in the neonatal medial prefrontal cortex (mPFC) on anxiety and depressive-like behavior in adolescents. Elevated expression of mature BDNF (LV-BDNF) increased anxiety and depressive-like behavior at P30. Only immobility in the tail suspension test was increased after expression of mutant BDNF (LV-pBDNF mut). Using our RNA-seq data and available online sn-RNAseq results, we investigated transcriptomic changes in the neonatal mPFC at P8 that underlie subsequent behavioral changes. Mature BDNF expression caused an increased transcriptional response in perivascular stromal cells (PSC) with such genes: Ptgds, Slc6a13, Slc22a6, Bnc2, Slc13a4, Aldh1a2. Based on GWAS data, Ptgds is a candidate gene associated with ADHD and bipolar disorder Pujol-Gualdo et al. (2021); Marín-Méndez et al. (2012); Munkholm et al. (2015). LV-pBDNF mut caused a complete opposite set of transcriptional changes in the PSC compared to LV-BDNF. The observed similar behavioral phenotype after expression of mature and mutant forms of BDNF together with the detected genes related to bipolar disorder underpinned that Bdnf could play a substantial role in the pathogenesis of this neurobehavioral disorder.

Introduction

Brain maturation is a complex and intricate process that takes considerable time to develop. Several shifts of main lead programs occurred during this multiplexing process (Zhong et al., 2018; Amiri et al., 2018; Polioudakis et al., 2019). First neurogenesis gives way to astro-, then oligodendrogenesis and synaptogenesis (Zhong et al., 2018; Amiri et al., 2018; Polioudakis et al., 2019). Changes in key processes are accompanied by changes in cell dynamics and transcriptomics. The early neonatal period is crucial for further brain development and mental abilities (Lanshakov et al., 2016). Harmful and stressful effects during these periods have long-lasting consequences on mental health and abilities. It has been brilliantly demonstrated in monocular deprivation experiments in the early postnatal period and the lifelong consequences of such effects (Fong et al., 2016; Pizzorusso et al., 2006). During the neonatal period, neural circuits are tuned and developed (Luhmann et al., 2016).

Maternal stress during pregnancy, adverse early life experiences may predispose individuals to develop psychopathology in adulthood (Satinsky et al., 2021; Xia et al., 2023). Another factor that affects the developing nervous system and is commonly encountered in perinatal medicine is antenatal glucocorticoid therapy, which is prescribed to women at risk of preterm delivery. We have previously shown that administration of the synthetic glucocorticoid dexamethasone causes neuronal apoptosis in the subiculum of neonatal rat pups (Lanshakov et al., 2016). Maternal behavior and early environmental conditions could affect offspring brain maturation through DNA methylation and epigenetic mehanisms (Weaver et al., 2004) involving Bdnf exon IV differential transcription in the prefrontal cortex (PFC) (Costa et al., 2023).

If we were to talk about genes and factors that orchestrate central nervous system (CNS) development and cross-talk with environmental factors, we should first talk about neurotrophic factors or Bdnf. Bdnf is a secreted neurotrophic factor with complex expression regulation and protein processing (Mowla et al., 2001; Menshanov, Lanshakov & Dygalo, 2015). First, its mRNA is translated into preproform, which is processed by cellular proteases into proform (Menshanov, Lanshakov & Dygalo, 2015). Furin and several other cell convertases then cleave the proform to mature BDNF. proBDNF and matBDNF mostly bind to different receptor types in the brain (Menshanov, Lanshakov & Dygalo, 2015). Proform binds to the p75NTR and sortilin/SorCS1-3 coreceptors (Glerup, Nykjaer & Vaegter, 2014; Glerup et al., 2014; Glerup et al., 2016; Kraemer et al., 2014; Leloup, Chataigner & Janssen, 2018), and mature form binds to TrkB–kinase receptor (Minichiello, 2009). p75NTR and TrkB have different receptor pathways and may have different effects on behavior in addition to their roles in neuronal plasticity and synapse formation (Glerup et al., 2016; Glerup, Nykjaer & Vaegter, 2014; Glerup et al., 2014; Kraemer et al., 2014; Leloup, Chataigner & Janssen, 2018; Minichiello, 2009; Marchetti et al., 2019; Rezgaoui et al., 2001).

We are far from a complete understanding of the precise molecular mechanisms of action of proBDNF and matureBDNF in the neonatal brain. Alteration of pro- and mature BDNF signaling during the early postnatal period could have a long-lasting outcome on mental health and predisposition to anxiety and depression in adulthood. Medial prefrontal cortex (mPFC) neuronal activity and BDNF signaling play key role in resisting depression (Hare & Duman, 2020). Opposite changes in BDNF expression are observed in the brains of patients with major depression. Upregulation of BDNF occurs in the amygdala and nucleus accumbens, while downregulation of BDNF occurs in the hippocampus and medial prefrontal cortex (mPFC) (Phillips, 2017). From a receptor perspective, experiments with TrkB and TrkBT1 have also had opposite results on anxiety and depression. Mice overexpressing TrkBT1 (i.e. reduced BDNF signaling) showed decreased exploratory activity (Kemppainen et al., 2012), whereas mice lacking TrkBT1 (i.e. increased BDNF signaling) showed increased anxiety (Watson et al., 2015). BDNF abnormalities also contribute to astrocyte and microglial dysfunction in depression pathways (Phillips, 2017). During the first postnatal week, Bdnf is expressed in the hippocampus and mPFC of rodents (Mowla et al., 2001; Baquet, Gorski & Jones, 2004).

It is hypothesized that proBDNF plays opposite roles to mature BDNF in neuronal functions. To address this question and the role of pro- and mature BDNF in the neonatal prefrontal cortex, we used several methods: viral transduction, RNA-seq, and behavioral assays. Using an all in one Tet-ON doxycycline (DOX)-inducible system, we temporally increased the expression of different BDNF forms, in particular: mature BDNF, proBDNF, and proBDNF mutant form, where the convertase cleavage site was mutated -126RVRR129 →-126AAAA129. It could not be processed by Furin because of this mutation. After expression activation during the first postnatal week, RNA sequencing was performed to assess transcriptomic changes caused by different BDNF forms. To correlate transcriptomic changes with behavioral changes following increased expression of different forms of BDNF in the neonatal mPFC, anxiety and depressive-like behaviors were examined at P30 in adolescents. The present paper shows the major target genes of BDNF and proBDNF signaling in the neonatal mPFC. Temporal alteration of BDNF signaling in the neonatal mPFC had a significant effect on anxiety and depressive-like behaviors in adolescence.

Materials and Methods

Constructs

To create an all-in-one TET-ON lentiviral vector (LV) specifically expressed in neurons, the hPGK promoter was replaced with the hSyn promoter in the plasmid pCW57-MCS1-P2A-MCS2(GFP). This was done using two-step cloning. pCW57-MCS1-P2A-MCS2(GFP) was a gift from Adam Karpf (Addgene plasmid 80924; http://n2t.net/addgene:80924; RRID:Addgene_80924). First, pCW57-MCS1-P2A-MCS2 (GFP) was cleaved with AsuNHI/FspI, then the 155bp fragment was ligated to the 7237bp fragment obtained after pCW57-MCS1-P2A-MCS2(GFP) was cleaved with AsuNHI/HindII. Second, in the result, promoterless construct human synapsin promoter (hSyn) was cloned with MluI/BamHI sites, resulting in pCW57-MCS1-P2A-MCS2-hSyn(GFP) construct (Shaburova & Lanshakov, 2021). The hSyn sequence was taken from the plasmid pAAV-hSyn-hChR2(H134R)-mCherry. pAAV-hSyn-hChR2(H134R)-mCherry was a gift from Karl Deisseroth (Addgene plasmid 26976; http://n2t.net/addgene:26976; RRID:Addgene_26976). The cDNAs encoding proBDNF were amplified from the cDNA library of rat P4 brain pups using the primers listed in Table S1. The PCR product was T/A cloned into the pAL-TA (Evrogen) vector and verified by Sanger sequencing. Using PCR directed mutagenesis and primers from Table S1, the cleavage site and HAtag mutations were introduced. All intermediates have been T/A cloned. Subsequently, AsuNHI and SalI sites were introduced by PCR and sequences encoding different forms of BDNF were cloned into pCW57-MCS1-P2A-MCS2-hSyn(GFP).

Animals

All animal procedures were approved by the Animal Care and Use Committee of the Institute of Cytology and Genetics (Protocol 151 dated 28.04.2023) and were performed in accordance with the European Communities Council Directive 63/2010/EU. Animals were obtained from the conventional animal facility of the Institute of Cytology and Genetics. Gestational Wistar rats were individually housed at 22–24 °C, 12-hour dark/12-hour light cycle, ad libitum. Birth date was considered P0. The litters were limited to a maximum of 8 male puppies per litter. The remainder were crossbred from other dams if there were less than 8 males in the litter. The same litter contained animals from all of the experimental groups in a randomized fashion. To distinguish the baby rats, they were marked with red paint (fucorcin) on the right front paw, right side, right rear paw, tail base, left rear paw, left side, left front paw, head, according to the numbers from 1 to 8. The tags were updated on a daily basis. On P24, the mothers were removed. After weaning, the animals were housed in groups of 4 per cage. All experimental samples were blindly number-coded for further processing.

Cell transfection and western blotting

CHO-K1 cells (obtained from the cell culture collection of the State Research Center of Virology and Biotechnology Vector, Novosibirsk, Russia) were cultured in DMEM/F12 medium supplemented with 10% FBS, 1X NEAA, 1 mM L-glutamine, 50 U/ml penicillin, and 50 mg/ml streptomycin. The day before transfection, CHO-K1 cells were seeded on a 48-well plate with a density of 3.5* 104 cells per cm2. The next day, cells were transfected with 0.5 µg of plasmid DNA per well and Lipofectamine 3000, dissolved in DMEM/F12, according to the manufacturer’s instructions. The next morning, a fresh growth medium was added. Half of all samples had DOX in the medium at a concentration of 20 ng/ml. After 1 hour of incubation, the medium was removed and the cells were washed once with warm 1xPBS. After that 200 µl RIPA lysis buffer (50 mM Tris pH 7.4, 0.1% SDS, 0.5% sodium deoxycholate, 1.0% NP-40, 1.0% Triton X-100, 150 mM NaCl) together with 1x Protease Inhibitor Cocktail (P2714, Sigma Aldrich) was added in each well and cells were scraped. For the proper lysis and dis-aggregation samples were passed through 10 strokes of insulin syringe with 29Ga needle. Cell debris was then removed by centrifuging at 12,000 g for 10 minutes at 4 ∘C. The Lowry assay was used to measure protein concentration. Lysates containing 20 µg of total protein were denatured at 95 ∘ for 5 min in 1x Laemli buffer (50 mM Tris-HCl (pH 6.8), 10% glycerol, 100 mM β-mercaptoethanol, 1% SDS, 0.002% bromophenol blue). They were then loaded onto an Any kD Mini-PROTEAN TGX Stain-Free Protein Gel (Bio-Rad, Hercules, CA, USA) and run at 100 V for 1 hour in running buffer (25 mM Tris, 192 mM glycine, 0.1% SDS, pH 8.3). The proteins were then transferred to 0.2 µm nitrocellulose membrane (Bio-Rad, Hercules, CA, USA) using the following parameters: 100 V, 20 min in transfer buffer (47.9 mM Tris, 38.6 mM glycine, 20% methanol, pH8.3). Membranes were blocked for 1 hour at RT in 0.5% fish skin gelatine 0.1% bovine casein in 1xPBS with 0.1% Triton X-100 (1xPBST) and incubated overnight at 4 ∘C with anti-HA-tag primary antibody (SAB4300603, Sigma-Aldrich, St. Louis, MO, USA) diluted 1:2,000 in Sea Block (37527, Thermo Fisher Scientific, Waltham, MA, USA). The next day, the membranes were washed in 1xPBS 0.1% SDS and incubated with secondary antibody (goat anti-rabbit IgG; Bio-Rad, Hercules, CA, USA) 1:5,000 diluted in blocking buffer for 1 hour at RT. SuperSignal West Femto Maximum Sensitivity Substrate (Thermo Fisher Scientific, Waltham, MA, USA) was used to amplify the chemiluminescence signal. The membranes were imaged using the Chemidoc Touch Imaging System (Bio-Rad, Hercules, CA, USA) and the staining intensity of the bands corresponding to the analyzed proteins was evaluated. The membranes were then stripped for 30 min at 50°C in harsh stripping buffer 2% SDS 62mM Tris HCl pH =6.8 0.8% ß-mercaptoethanol, reprobed overnight for BDNF staining (ab6201, Abcam, Cambridge, UK), stripped again reprobed for VP16 (ab4808, Abcam, Cambridge, UK), washed and probed for actin (ab179467, Abcam, Cambridge, UK)

Immunohistochemistry

Immunohistochemistry procedure was done as previously described Lanshakov et al. (2016), Lanshakov et al. (2021). After lentivirus infusion on P1 (LV-Syn,n = 2, 18 sections per animal), rat pups were deeply anesthetized with Avertin and transcardially perfused with 1xPBS followed by 1xPBS buffered 4% PFA on P8. The brains were postfixed for 4 hours in 1x PBS buffered with 4% PFA. Then cryoprotected first in 15% sucrose in 1xPBS overnight followed by incubated in 30% sucrose overnight. According to the Neonatal Rat Brain Atlas (Khazipov et al., 2015), 25 µm frozen tissue sections were prepared around the mPFC region using the MICROM350 cryostat. Sections were placed on SuperFrost Plus slides (Thermo Fisher Scientific, Waltham, MA, USA). Sections were washed in 1xPBST 0.2% Triton X-100. Blocking was performed with Sea Block (37527, Thermo Fisher Scientific, Waltham, MA, USA) for 1 hour at RT. Sections were then incubated overnight at 4 °C with primary antibodies: rabbit anti-IBA1 (ab178846, Abcam, Cambridge, UK) or rabbit anti-GFAP (Z0334, Dako), and mouse anti-SATB2 (sc-81376, Santa-Cruz), all diluted 1:200 in Sea Block (37527, Thermo Fisher Scientific, Waltham, MA, USA). The next day, sections were washed three times for 15 minutes in 1xPBST. Sections were then incubated with F(ab)2 donkey anti-rabbit Alexa 647 (Jackson Immunoresearch, West Grove, PA, USA), and F(ab)2 donkey anti-mouse Cy3 (Jackson Immunoresearch, West Grove, PA, USA) secondary antibodies diluted 1:500 in Sea Block for 1 hour at room temperature. The slides were then washed three times for 10 minutes each time in 1x PBST. Coverslips were mounted with Mowiol 4-88 (81381, Sigma Aldrich, St. Louis, MO, USA) containing nuclear counterstain DAPI. Confocal laser scanning microscopy (CLSM, Zeiss LSM 510) was used to analyze the sections. The following laser lines were used for CLSM imaging: DAPI–405 nm, Alexa 488–488 nm, Cy3–543 nm, Alexa 647–633. A tile-scan function (Plan-Apochromat 20x/0.8 M27 objective and 200 µm pinhole) was used to generate panoramic images of the mPFC region. The number of double and triple positive cells was evaluated using the open source software QuPath. Briefly, the EGFP channel was used to detect and mask cells. An appropriate classifier was then created for each channel based on the mean cytoplasmic signal of the detected cells for IBA1 and GFAP or the mean nuclear signal for SATB2 (Santa Cruz). QuPath’s definition of double positives is cells that meet the requirements of two classifiers. See the QuPath documentation for a detailed description (QuPath, 2024).

Animal experimental design

For the temporal expression increase of different BDNF forms specifically in neurons of the neonatal mPFC, all were cloned in a TET-on lentiviral construct (Fig. 1A). The furin cleavage site was replaced by 4 alanines (126RVRR129 → -126AAAA129). This mutant form (LV-proBDNF_mut) could not be processed by cell proteases. The mature form of BDNF (LV-BDNF) was cloned without the propeptide coding portion. ProBDNF (LV-proBDNF) had no changes except for the C-terminal tag. All forms of BDNF had an HA tag at the C-terminus. At P1 (Fig. 1B), lentiviral particles were infused. The elevated expression of BDNF forms was done from P3 to P8 with a daily subcutaneous injection of DOX 0.2 mg/kg in 100 µl saline. The pups were then returned to their mothers (eight pups per cage, 12 animals per group for behavioral assay, three animals per group for RNAseq) in the home cage and kept intact until P30 for behavioral testing. Animals were then euthanized with CO2. A portion of the animals were sacrificed at P8 (three animals per group) for transcriptomic study.

Figure 1 Experiments main scheme.

(A) Diagram of TET-ON Lentivirus (LV) constructs. (B) LV particles were injected into the mPFC area on P1. DOX for expression induction was injected subcutaneously on P3-P8 and behavioral testing was performed on P30. (C) Sequenogram of the cleavage site in proBDNF and mutant constructs (D) Representative Western blot image showing induction of construct expression in CHO-K1 cells. (E) Panoramic CLSM image of the P8 rat pup brain after LV-Syn injection on P1 (F) Results of immunohistochemical quantification SATB2/GFAP double immunohistochemistry (G) Results of immunohistochemical quantification SATB2/IBA1 double immunohistochemistry (H) Representative higher magnification microscopic images of mPFC (orthoprojections).

Lentivirus production

Lentiviruses were produced in HEK293FT cells (R70007, Thermo), PEG-precipitated, chromatographically purified on quaternary ammonium columns according to Shaburova & Lanshakov (2021). qPCR was used to determine lentiviral titers. Only the 109 titer preparation was injected into the mPFC.

Stereotaxic lentivirus injection to rat P1 mPFC

According to R.Z. Khazipov’s Neonatal Rat Stereotaxic Atlas (Khazipov et al., 2015), lentiviral particles were stereotaxically infused into P1 rat pups. At P1, rat pups were anesthetized with ice cold. They were placed in ice for 5 minutes wrapped in a glove. Hypothermia was chosen because there was less bleeding after the infusion. The pups were then placed in rat stereotaxic apparatus with neonatal adaptors (J & K Seiko, China). One microliter of virus was infused into both hemispheres for 7 minutes using a 10 µl Neuros syringe (Hamilton, Reno, NV, USA) and a 30Ga needle (Hamilton) and a precision syringe pump(LongerPump, China) according to the following coordinates: AP +1.6 mm, ML +0.8 mm, DV +2 mm. After infusion, the needle was left in place for another 7 minutes. A 30Ga Hamilton needle was used. The puncture was made directly to the head. After surgery, 100 µl of 100 µg/mL penicillin and 100 µg/mL streptomycin solution in 1xPBS were injected intraperitoneally. The pups were then recovered on the warming pad and returned to the mother.

Behavior testing

At P30, anxiety-like behavior was assessed using the light-dark box (LDB) test. Rats were tested in the (30 cm × 60 cm × 30 cm) halved into two compartments: light and dark (30 cm × 30 cm × 30 cm). All the walls in each part were either white or black. The partition connecting two compartments had an opening of 6.5 cm × 6.5 cm. The light in the room was dimmed. A small 5W gooseneck lamp with a clamp was mounted above the light compartment, providing light (600 lux) to the center of the box. Illumination in the dark compartment was 5 lux (Narayanan & Kumar, 2018). Digital camera captures video above the device. Animals were placed in the light compartment facing the door between compartments. The duration of the test was 5 minutes. After each animal, the apparatus was cleaned with 70% ethanol and allowed to dry before the next subject was tested. After testing, the animals were placed in a separate cage and returned to the home cage when all animals in that cage had been tested. In the LDB test, the time in the white field, the number of complete transitions, and the number of stretches from the dark field to the white field were counted on videotapes by three independent observers. One week after the first test, depressive-like behavior was assessed with the Tail Suspension Test (TST) (Can et al., 2011). The animals were suspended 60 cm off the ground with paper tape over medical tape attached to a rat’s tail. Apparatus were isolated from sides and back. Test lasted 5 minutes as LDB. After each animal, the arena was cleaned with 70% ethanol. After the test, the animals were placed in a separate cage and returned to the home cage when all animals in that cage were tested as above. In the TST, total immobile time, active time, and latency were counted by three independent observers.

Statistics

Statistical analyses were performed with R. For the behavioral analysis, linear mixed effects models (nlme package (José C. Pinheiro, 2000)) were used with lentiviral injection group as a fixed effect and animal ID as a random effect. Mixed models are often preferred to traditional analysis of variance and regression models because of their flexibility in dealing with missing values and uneven distribution of a random variable (Yang et al., 2014). Means were compared by Tukey test, emmeans R package (Searle, Speed & Milliken, 1980). Normality was tested using the Shapiro–Wilk test. Data of time spent in the white compartment were Boxcox transformed (Daimon, 2011). Immunohistochemistry data were analyzed using the non-parametric Mann–Whitney test.

mPFC samples collection and RNA extraction

Pups were sacrificed by rapid decapitation with scissors one hour after the last DOX injection at P8. There were three animals per group for the RNAseq experiments. To enrich RNA samples for neuronal transcripts, mPFC was dissected under a binocular microscope and the pia was removed. The samples were then immediately weighted. The total weight of each mPFC sample did not exceed 26 mg. RNA was isolated using the Norgen Total RNA Purification Kit according to the manufacturer’s protocol. RNA samples were treated on columns with RNAse-free DNAse I (Sigma Aldrich). After elution, 40 U of ribonuclease inhibitor Ribolock (Thermo) was added to the samples.

RNA-sequencing

RNA samples were used to prepare cDNA libraries using the QIAseq Stranded Total RNA Lib Kit (Qiagen, Hilden, Germany) according to the manufacturer’s protocol. Ribosomal RNA depletion was performed with NEBNext rRNA Depletion Kit (New England Biolabs Inc., Ipswich, MA, USA). The concentration of cDNA libraries was measured by the dsDNA High Sensitivity Kit on a Qubit 4.0 fluorometer (Thermo Fisher Scientific). The quality of cDNA libraries was assessed using High Sensitivity D1000 ScreenTape on a 4150 TapeStation (Agilent). Libraries were sequenced on a NextSeq instrument (Illumina) using single-end 75 bp reads, 15M reads per sample. Each experimental point was biological replicate. The statistical power of this experimental design, calculated in RNASeqPower is 0.8. Adapter sequences were cut with Trimmomatic (Bolger, Lohse & Usadel, 2014). Raw reads were aligned to the mRn7.2 reference genome using Hisat2 (Kim, Langmead & Salzberg, 2015). The number of mapped reads was counted by the Rsubread (Liao, Smyth & Shi, 2019). The edger package (McCarthy, Chen & Smyth, 2012) was used to detect DEGs between LV-BDNF vs LV-Syn; LV-proBDNF vs LV-Syn; LV-proBDNF_mut vs LV-Syn. Edger’s default Trimmed Mean of M (TMM) method was used to normalize the libraries. Edger’s Generalized Linear Models (GLM) approach was used to identify the DEGs.

Functional annotation

Functional annotation of differentially expressed genes was performed using the R clusterprofile package (Wu et al., 2021). DEGs with |log2FC| > 1 and p < 0.05 have been selected.

snRNAseq data

Single-nucleus RNAseq data from neonatal rat cortex were obtained from the control samples of the Chen et al. (2022) study (NCBI GEO: GSE185538). The Seurat 4.3.0 package was used to analyze the gene expression of cell clusters (Hao et al., 2021)

Results

Proof of constructs

Based on Dr. Adam Karpf’s plasmid, we have constructed an all-in-one TET-ON lentiviral construct for transgene expression induction with DOX in neurons. Sequences encoding pro-, mature and mutant Bdnf with HA tag were then placed under the tight-TRE promoter (Fig. 1A) and verified by Sanger sequencing (Fig. 1C). After stereotaxic lentiviral infusion to P1 male rat mPFC expression was induced for 6 consecutive days from P3 to P8 (Fig. 1B) for the main experiment. See the Methods section for a detailed description. For the initial evaluation of the construct, we performed preliminary experiments with transfection of CHO-K1 cells and Western blotting. These experiments showed the absence of protein products without DOX and the presence of all expected forms 14kDA for mature BDNF and 28kDA for proBDNF and proBDNF_mut with anti-HA tag antibodies when DOX was added to the cell culture medium (Fig. 1D, original Western blot photos are in the Supplementary Material). After stripping, the membrane was reprobed with anti-BDNF antibody. Bands corresponding to proBDNF were present in all lines. After another round of stripping, the membrane was probed for VP16, which is present in rtTA. All lines had corresponding bands. We then checked which cells in the neonatal mPFC we could see expressing the EGFP reporter. Stereotaxically infused control LV-Syn lentivirus on P1, we looked at reporter expression and injection site placement on P8 to fit the scheme of this experiment with the main RNA-seq experiment. Panoramic confocal images show that with the chosen stereotaxic coordinates we obtained reporter expression in the mPFC (Fig. 1E). Double immunohistochemistry with callosal neuron marker SATB2 and mikroglial marker IBA1 using EGFP fluorescence showed that approximately 80% (95% CI [80.5–84.7]) of EGFP-positive cells were SATB2-positive and only 0.03% (95% CI [−0.0123–0.0679]) were IBA1-positive (p.adj(BH) < 0.01, Mann-Whitney U test, Figs. 1F, 1H. A slightly increased 6.19% (95% CI [4.35–8.03]) number of triple positive cells was observed. This could be explained by the expression of SATB1 in the immune cells and weak antibody cross-reactivity (Burute, Gottimukkala & Galande, 2012). Double immunohistochemistry with SATB2 and the astrocyte marker GFAP again showed about 80% EGFP/SATB2 positive cells (96% CI [78.0–83.3]) and almost no double EGFP/GFAP 0.1% (95% CI [−0.0344–0.234]) and triple positives 0.694% (95% CI [0.221–1.17]; p.adj(BH) < 0.01, Mann-Whitney U test, Figs. 1G, 1H). This means that most of the transduced cells were neurons.

Increased expression of different forms of BDNF in the mPFC in the neonatal period has an effect on anxiety and depressive-like behavior in the adolescent period

Temporary increase in expression of mature BDNF (LV-BDNF) in the mPFC had a significant effect on anxiety (Fig. 2). Induction of LV-BDNF expression in the neonatal period had an anxiogenic effect in adolescence. These animals spent significantly less time in the white compartment (p = 0.02, AIC = −33.12748 BIC = −21.90027) compared to the control LV-Syn. No significant changes in anxiety were observed after expression induction of other constructs. The latency to enter the dark compartment was also not significantly changed in any group. There were no significant differences between groups for other parameters. We were also surprised by changes in depressive-like behavior. The immobility time in the tail suspension test was significantly higher in the LV-BDNF group with induction of mature BDNF expression at P3-P8 (p = 0.0327, AIC = 473.5392, BIC = 484.7664) compared to the LV-Syn control. Increased expression of the mutant form of LV-proBDNF_mut also caused an increase in immobility time (p = 0.02472) (Fig. 2) compared to control LV-Syn. Latency to immobility did not change between groups. Thus, we could conclude that increased expression of the mature form of BDNF in the neonatal mPFC is involved in increased anxiety and depressive-like behavior in adolescence. Expression of the mutant form has the same effect, but only on depressive-like behavior.

Figure 2 Behavior changes at P30.

Upper graph row - LDB test results, lower row - TST results.

Transcriptomic changes at P8 following lentiviral expression induction

We saw behavioral effects only after activation of LV-BDNF and LV-proBDNF_mut expression, but the transcriptome was altered in each group. In the PCA plot, the distribution of experimental points in the PCA dimensions depends on the observed total transcriptomic changes (Fig. 3). Interestingly, experimental points of animals with proform expression (LV-proBDNF, LV-proBDNF_mut) were closer to controls (LV-Syn, (Fig. 3)), more distinct points were with expression of mature BDNF. The volcano plots show that genes that are upregulated in the LV-BDNF group are downregulated in LV-proBDNF_mut or LV-proBDNF. These genes are Ptgds, Fam180a, Itgb4, Ranbp3l, Bnc2, Slc6a13, Slc22a6 (Fig. 4, Table S1). There were also genes that were present in all three DEG lists after viral injections. They were Ptprh, which was upregulated, and Ifitm3, which was downregulated. Amount of common genes represented on a Venn diagram.

Figure 3 RNAseq experimental points displayed in principal component analysis (PCA) dimensions and variability described by these components.

Figure 4 (A) Volcano plots of selected DEGs after each LV injection and expression induction.

Genes with |log2FC| > 1 and p < 0.05 were selected. (B) Venn diagram of common genes of DEGs.

Functional annotation of differentially expressed genes

Surprisingly, GO enrichment analysis yielded more terms only after LV-BDNF and far fewer terms after LV-proBDNF and LV-proBDNF_mut (Fig. 5). After BDNF expression induction were enriched such terms-“bone development”, “endochondral bone morphogenesis”, “bone morphogenesis”, “cartilage development”, “ossification”, “response to retinoic acid”, “extracellular matrix organization”, “response to estradiol”, “positive regulation of hormone biosynthesis process”, “transforming growth factor beta 1 production”. After LV-proBDNF, “rough endoplasmic reticulum”, “basolateral plasma membrane”, “nuclear membrane”, “nuclear envelope lumen”, “outer dense fiber” were enriched. After LV-proBDNF_mut, the list of terms was completely different: “extracellular matrix structural component conferring tensile strength”, “extracellular matrix structural component”, “SMAD binding”, “rDNA binding”, “phospholipase A1 activity”, “oxidoreductase activity, acting on NAD(P)H, heme protein as acceptor”, “co-SMAD binding”. If we look at the cnetplots (Fig. 5) of the enriched terms and the related genes that fulfill these terms after each viral injection, it is mostly the genes mentioned above (Bnc2, Ptgds, Rnabp3l, Slc6a13, Slc22a6, Slc13a4, Tgif1) and shown in the second column of plots in the figure. Rarb, Igf2, Bmp6, Penk, Clec3b, Aldh1a2, Slc13a4 were upregulated only after LV-BDNF. Slc6a13, Slc22a6 from the term “basolateral plasma membrane” were upregulated after LV-BDNF and downregulated after proBDNF.

Figure 5 Enrichment analysis of DEGs on the basis of clusterProfiler packages.

Major biological processes GO terms of |log2FC| > 1 - left panel. Gene concept network (cnetplot) of the most enriched BP GO terms and the overlapping set of genes consisting of these terms after each LV injection - right panel.

Neonatal neocortex sn-RNAseq

We took only control samples from Chen’s study (Chen et al., 2022). The analysis in the Seurat package revealed a slightly larger number of cell types. However, the overall picture remains the same (Fig. 6) Slightly different parameters selected for clustering could explain additional cell cluster.

Figure 6 Results of snRNAseq data analysis and clustering of cells.

Interestingly, there are three types of astroglial cells in the rat neonatal cortex, and only one of them expresses GFAP. Cells of two other astroglial clusters express Aqp4 and Slc1a2 (Fig. 7). Although it is not correlated for the present study, we consider this fact very important for developmental neuroscience, together with the consistency validation of our analysis of snRNAseq data.

Figure 7 FeaturePlots showing the expression of selected genes Aqp4, Slc1a2, Gfap in three classes of astrocytes in neonatal rat brain.

logFC scale shown in purple.

Cell specificity of DEG expression

Analysis of selected DEGs described above in the context of rat neonatal cortex cell types revealed that these genes are predominantly expressed in clusters 16 and 17. It could be seen on the feature plots of the selected genes (Fig. 8). The list of genes that are exclusive markers for these clusters is shown in Tables S2–S3. In cluster 16, the perivascular stromal cells (PSC), Col1a1, Bnc2 are expressed. Cluster 17 consists of pericytes (PC) that express Pdgfrb, Vtn, Ptgds (Tables S2–S3, Figs. S2–S3). As mentioned above, in these (PSC, PC) cell types, the expression of mature BDNF caused an increase in the expression of marker genes. Expression of proBDNF and its mutant form decreased the mRNA level of these genes. This means that after expression activation, the main transcriptional response was observed in PSC and PC.

Figure 8 FeaturePlots showing the expression of selected genes that were mainly changed after BDNF expression induction across cell clusters.

logFC scale shown in purple.

Predominantly neuronal and glial genes altered after expression induction

As shown earlier, we have seen mostly transcriptional responses from PSC and PC cells. It was opposite depending on mature or proform expression induction. At the same time, changes in anxiety and depressive-like behaviors were the same. To further elucidate the molecular mechanism underlying these processes, we searched the DEG lists for genes that are predominantly expressed in neurons and glial cells (Fig. 9). After LV-BDNF we should note Naaladl2 log2FC = 1, Ifitm2 log2FC = 1, Atp5mk log2FC = −1.72, Rarb log2FC = 1.5, Fam171a1 log2FC = −1.37, Tgif1 log2FC = 1.43, Nmi log2FC = 1.11, Penk log2FC = 1.25. After LV-proBDNF Styk1 log2FC = 1.4, Rps21 log2FC = −1.03, Nr4a2 log2FC = 2.62, Tex47 log2FC = −1, Orai1 log2FC = 1.13, Fam171a1 log2FC = −1.23, Rsph1 log2FC = −1.03, Zfp52 log2FC = 1.03, Susd2 log2FC = 1.23, Pnpla1 log2FC = −1.03, Fcgr3a log2FC = 1.23, Tmco4 log2FC = −1.65, Traf5 log2FC = 1.33 were changed. After LV-proBDNF_mut Snrpf log2FC = −1.13, Etfb log2FC = −1.09, Dll3 log2FC = −1.17, Tex47 log2FC = −1.09, Rbm46 log2FC = −1.16, RGD1359290 log2FC = −1.01, Sim2 log2FC = −1.06, Susd2 log2FC = 1.33, Grm4 log2FC = 1.03, Fa2 h log2FC = −1.49, Gadd45b log2FC = −1.11, Fam171a1 log2FC = −1.5. All these genes are mainly expressed in different neuronal and glial cell types, with the exception of Fcgr3a, which is expressed in microglia. After all three LV injections, Fam171a1 was downregulated. After Proforms expression induction, Susd2 and Tex47 changed in a co-directional way. The other genes were exclusive to each LV.

Figure 9 FeaturePlots showing mainly neuronal and glial genes that were affected after BDNF expression induction.

logFC scale shown in purple.

Discussion

Temporally increased expression of pro-, mature-, and unprocessed (mutant) BDNF forms in the neonatal mPFC had an effect on anxiety and depressive-like behaviors in adolescence. We expected an anxiolytic and antidepressant effect. Instead, we got the opposite. Expression of mature BDNF increased indicators of anxiety and depressive-like behavior in the tests, expression of mutant BDNF (proBDNF_mut) increased only depressive-like behavior. Behavior was not affected by expression of proBDNF.

Mature BDNF is known to improve memory and cognition through TrkB receptors (Cunha, Brambilla & Thomas, 2010; Miranda et al., 2019). Individuals with higher cognitive abilities have higher demands on environmental conditions. Therefore, increased anxiety and depressive-like behavior may be related to our observations. To summarize, we can say that people with a higher IQ tend to be predisposed to anxiety and depression (Melby et al., 2020). Our findings are also supported by the fact that mice lacking Trkbt1, with increased mature BDNF signaling, showed increased anxiety (Watson et al., 2015). To determine which gene expression changes in the mPFC at P8 are responsible for behavioral changes, we performed RNA-seq analysis of mPFC samples. Surprisingly, we saw the main transcriptional response in PSC and PC cells. Perivascular stromal cells in the brain are a heterogeneous population of cells associated with the vasculature, including endothelial cells, pericytes, and astrocytes. These cells contribute significantly to the structure, function, and integrity of the blood-brain barrier (BBB) and are involved in various processes such as vascular architecture, scavenging and immunoregulatory functions (Kida et al., 1993), maturation, and neuroinflammation, mesenchymal and neuroectodermal differentiation potential (Paul et al., 2012; Lojewski et al., 2015). Brain pericytes are multifunctional cells located on the abluminal side of the capillaries throughout the brain, playing pivotal roles in the maintenance and function of the neurovascular unit(NVU) (Brown et al., 2019). These cells are crucial for the development and integrity of the blood-brain barrier (BBB), regulation of cerebral blood flow, vascular development and angiogenesis, neuroinflammation management (Rustenhoven et al., 2017), and even stem cell-like activities (Dore-Duffy, 2008).

The observed transcriptional response was inversely dependent on the form of BDNF, with marker gene expression upregulation after mature BDNF and downregulation after proforms. We could not conclude that this expression reflected changes in the amount of a particular cell type. Detailed immunohistochemical studies using cell type antibody markers are required to make such a conclusion.

Notable genes that behave in this way are Bnc-2 Basonuclin 2 (PSC) and Ptgds Prostaglandin D2 Synthase (PC). Ptgds is highly expressed in the PC then in the PSC log2FC 5.5 vs. 2.78. Some weak expression is detected in neuronal and glial cells (Fig. 8). It has been shown that in patients with rapid-cycling bipolar disorder, Ptgds mRNA levels are reduced in peripheral blood mononuclear cells. Prostaglandin D2 synthase (PTGDS) is an enzyme that catalyzes the conversion of prostaglandin H2 (PGH2) to prostaglandin D2 (PGD2). PGH2 is a common precursor for various prostaglandins, which are lipid compounds that participate in a wide range of body functions including inflammation, pain, and fever responses, as well as the regulation of blood pressure and coagulation. PTGDS specifically takes PGH2 and converts it into PGD2, a prostaglandin involved in several important physiological processes: sleep regulation (Pahl, 2007), inflammation and immune response (Rajakariar et al., 2007), allergic reactions (García-Solaesa et al., 2014), vasodilation and blood pressure regulation (Mohri et al., 2007), platelet aggregation. PGD2 is a major prostaglandin produced in the central nervous system and is involved in the regulation of sleep and pain responses through DP receptors. It acts as a sleep-promoting substance (Pahl, 2007).

In the immune cells prostaglandin D2 synthase (hPGD2S) metabolizes cyclooxygenase (COX)-derived PGH2 to PGD2 and 15-deoxy δ12–14 PGJ2. PGD2 plays crucial roles in the control of inflammation, where it can act as both a pro- and anti-inflammatory mediator (Rajakariar et al., 2007). PGD2’s role in vasodilation and the regulation of blood pressure is significant, illustrating the compound’s effect on vascular diameter and cardiovascular health. This function is critical for maintaining vascular homeostasis (Mohri et al., 2007). Genome-wide association study of sporadic brain arteriovenous malformations showed Bnc2 intron SNP association with nominal p < 0.1 (Weinsheimer et al., 2016). Experiments on BV2 cell lines suggest that circular RNA Circ-Bnc2 alleviates neuroinflammation in LPS-stimulated microglial cells through regulating miR-497a-5p/HECTD1 axis (Chen & Cao, 2023). Circular RNAs (circRNAs) like circ-Bnc2 are generated from pre-mRNA back-splicing, where a downstream splice donor site is joined to an upstream splice acceptor site. This process leads to the production of a covalently closed loop structure without 5′to 3′polarity and a polyadenylated tail, distinguishing them from linear RNA counterparts.

Slc6a13, Slc22a6, Slc13a4 is another set of genes that should be noted. These genes consist of the solute carrier family. While Slc6a13 shares the solute carrier family designation with Slc6a1, indicating it’s part of the broader family of membrane transport proteins that play crucial roles in neurotransmitter regulation, the specific details regarding Slc6a13 and its link to neuropsychiatric disorders are uncertain. Slc6a13 is known to encode a GABA transporter, suggesting its importance in GABAergic signaling pathways similar to Slc6a1 (Bhatt et al., 2023). The Slc6a1 gene has been linked to various neurodevelopmental disorders (Goodspeed et al., 2023), including epilepsy (Johannesen et al., 2023), intellectual disability(ID), and autistic spectrum disorders (ASD), among others. The gene encodes for the GABA transporter 1 (Gat-1), which is responsible for the reuptake of GABA from the synaptic cleft, playing a crucial role in maintaining neurotransmitter homeostasis and modulating neurotransmission. Abnormalities in Gat-1 function due to Slc6a1 variants can disrupt GABAergic signaling, contributing to the pathogenesis of these disorders.

Slc22a6, known as the organic anion transporter 1 (OAT1), is part of the SLC22 family, which includes organic cation transporters (OCTs), organic anion transporters (OATs), and organic zwitterion/cation transporters(OCTNs) (Parker et al., 2023) . This family plays a crucial role in regulating the cellular uptake and excretion of various endogenous and exogenous substances, including metabolic by-products, toxins, and pharmaceuticals. While primarily recognized for their roles in the kidneys and liver, members of the SLC22 family also have significant functions in the brain, contributing to homeostasis and the detoxification of harmful substances. Slc13a4 is recognized for its role in transporting nutrient sulfate across the placenta to the fetus, which is crucial for human development (Rakoczy et al., 2015) However, its specific function in the brain is not well-documented in the studies available. It was shown that expression levels for the sulfate transporter Slc13a4 to be elevated during postnatal development, and sulfate accumulation in the brains of Slc13a4 +/- mice is reduced, suggesting a role for this transporter during this critical window of brain development (Harvey et al., 2021).

Observed similar behavioral changes after mature and pro-form expression induction led us to look deeper beyond the transcriptional response in PSC and PC cells. Genes primarily expressed in neurons and glial cells were differentially altered after each LV, except for Fam171a1, Susd2 and Tex47. Fam171a1, Astroprincin (Apcn) is a recently characterized transmembrane glycoprotein that is abundant in brain astrocytes and is overexpressed in some tumors (Wahab et al., 2020). However, detailed functional roles and mechanisms of Apcn in the brain and its contribution to diseases, including its overexpression in tumors, are still being elucidated. Astroprincin’s identification as abundant in astrocytes hints at its potential involvement in neurobiological processes and astrocyte functions. Astrocytes play crucial roles in maintaining the homeostasis of the central nervous system (CNS), including neurotransmitter uptake and recycling, providing metabolic support to neurons, contributing to the blood-brain barrier’s integrity, and participating in the brain’s response to injury. Susd2 regulates neurite outgrowth and synaptic density in hippocampal cultures. It is a transmembrane protein with a complement control protein (CCP) domain, previously identified as a tumor-reversing protein but without a characterized function in the CNS. Susd2 expression peaks two weeks after birth, coinciding with synaptogenesis (Nadjar et al., 2015). It localizes to soma, axons, dendrites, and associates with excitatory synapses. Inhibition of Susd2 reduces the number of excitatory synaptic profiles and alters morphological parameters, suggesting that it alters adhesion properties and plays a dual role at different stages of development. There is not much information about Tex47 function in the brain, it has primarily function in reproductive tissues such as the testes, influencing spermatogenesis or other aspects of male fertility..

Another gene of note that has been altered by LV-BDNF is the Rarb gene. Retinoic Acid Receptor Beta (RARB) plays a crucial role in various developmental and physiological processes, especially in the brain, mediated by retinoic acid signaling. Research has shown that RARB, along with other retinoic acid receptors, is involved in the molecular patterning of the PFC and motor areas, the development of reciprocal PFC-mediodorsal thalamus connectivity, and intra-PFC synaptogenesis. These processes are crucial for the proper function and structural development of the PFC, highlighting the importance of retinoic acid signaling in brain development and potentially its evolutionary expansion. Retinoic acid (RA) signaling, mediated through RARB and other receptors, is essential for the molecular patterning of the PFC. This signaling is required for the expression of layer 4 markers, intra-PFC synaptogenesis, and the development of reciprocal connectivity between the PFC and the mediodorsal thalamus. These findings indicate a critical role for RA signaling in the establishment of the structural and functional architecture of the PFC during development (Shibata et al., 2021).

Conclusions

In conclusion, we found that increased expression of mature BDNF (LV-BDNF) at P5-P8 resulted in increased anxiety and depressive-like behaviors at P30. Conversely, expression of a mutant form of BDNF (LV-pBDNF_mut) only increased immobility in the tail suspension test. Using our RNA-seq data and available online single-nucleus RNA sequencing results, we analyzed transcriptomic changes in the neonatal mPFC at P8 that may underlie the observed behavioral changes from a cell type perspective. Expression of matBDNF triggered an enhanced transcriptional response in perivascular stromal cells (PSC) and pericyte cells (PC), particularly affecting genes such as Ptgds, Slc6a13, Slc22a6, Bnc2, Slc13a4 and Aldh1a2. Interestingly, Ptgds is identified by GWAS as a candidate gene associated with ADHD and bipolar disorder (Pujol-Gualdo et al., 2021; Marín-Méndez et al., 2012; Munkholm et al., 2015). The observed similar behavioral phenotype after expression of mature and mutant forms of BDNF, together with the detected genes associated with bipolar disorder, supported that Bdnf could play a substantial role in the pathogenesis of this neurobehavioral disorder. It is noteworthy that proforms induced transcriptional responses in PSC, PC that were completely opposite to those induced by LV-BDNF. This study underscores the intricate role of BDNF’s various forms in the development of the nervous system and their potential long-term impacts on behavior, offering new insights into the neurobiological underpinnings of psychiatric disorders.

Supplemental Information

Supplemental Information 1 RNASeq results: Differential Expressed Genes

Supplemental Information 2 List of Selective Marker Genes of Cell Cluster 16

Supplemental Information 3 List of Selective Marker Genes of Cell Cluster 17

Supplemental Information 4 Immunohistochemistry data

SATB2/IBA1 Double IHC Staining

Supplemental Information 5 Double immunohistochemistry data

GFAP/SATB2 double IHC data

Supplemental Information 6 Tail Suspension Test dataset

Each data point indicate measured behavioral parameter for one animal

Supplemental Information 7 Dataset of light dark box test parameters results

Each data point indicate behavioral parameters for one animal

Supplemental Information 8 Arrive Checklist

Supplemental Information 9 Supplementary materials with original western blot pictures and primer sequences

RNA sequencing was carried out using the equipment of the Core Facility “Medical Genomics” (Tomsk NRMC) and the Tomsk Regional Common Use Center. Confocal images were acquired at the Institute of Cytology and Genetics Microscopy Facility. Bioinformatics analysis was carried out at the Information and Computing Center of Novosibirsk State University.

Additional Information and Declarations

Competing Interests

Author Contributions

Animal Ethics

DNA Deposition

Data Availability

The authors declare there are no competing interests.

Dmitriy Lanshakov conceived and designed the experiments, performed the experiments, analyzed the data, prepared figures and/or tables, and approved the final draft.

Elizaveta Shaburova performed the experiments, analyzed the data, authored or reviewed drafts of the article, and approved the final draft.

Ekaterina Sukhareva performed the experiments, analyzed the data, authored or reviewed drafts of the article, and approved the final draft.

Veta Bulygina performed the experiments, analyzed the data, authored or reviewed drafts of the article, and approved the final draft.

Uliana Drozd performed the experiments, analyzed the data, prepared figures and/or tables, and approved the final draft.

Irina Larionova performed the experiments, analyzed the data, prepared figures and/or tables, and approved the final draft.

Tatiana Gerashchenko performed the experiments, analyzed the data, prepared figures and/or tables, authored or reviewed drafts of the article, and approved the final draft.

Tatiana Shnaider performed the experiments, analyzed the data, prepared figures and/or tables, and approved the final draft.

Evgeny V. Denisov conceived and designed the experiments, analyzed the data, prepared figures and/or tables, and approved the final draft.

Tatyana Kalinina conceived and designed the experiments, performed the experiments, analyzed the data, prepared figures and/or tables, and approved the final draft.

The following information was supplied relating to ethical approvals (i.e., approving body and any reference numbers):

All animal procedures were approved through the Institute of Cytology and Genetics Animal Care and Use Committee (protocol 151 from 28.04.2023)

The following information was supplied regarding the deposition of DNA sequences:

The raw reads are available at ENA database: PRJEB47241.

The following information was supplied regarding data availability:

The behavior testing data, Immunohistochemistry data, supplemental figures and raw western blot photos are available in the Supplemental Files and also at Zenodo DOI: Lanshakov, D., Shaburova, E., Sukhareva, E., Bulygina, V., Drozd, U., Larionova, L., Gerashchenko, T., Shnaider, T., Denisov, E., & Kalinina, T. (2024). PeerJ Supplemet 2024 [Data set]. Zenodo. https://doi.org/10.5281/zenodo.13773297.

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
