# Peer review of "Transiently elevated expression of different forms of brain-derived neurotrophic factor in the neonatal medial prefrontal cortex affected anxiety and depressive-like behaviors in adolescence"

_PeerJ, doi:10.7717/peerj.18465_

## Round 0.1 · original submission · Major Revisions

Two of the reviewers requested major changes, and one reviewer only minor changes. The reviewers are making valuable comments to improve the paper. Please address all their concerns, and indicate how you dealt with all of them in the rebuttal letter.

·

Basic reporting

This study investigated the effect of transiently over-expression of BDNF in mPFC in depression/anxiety-related behaviors. There are some interesting findings, but still many drawbacks exists.

Experimental design

1.Why the timepoint of RAN seq and the behavioral tests was different? If so, how the genes changes can explain the behavioral differences?

Validity of the findings

1.In Figure1D, the expression validity of several BDNFs shoud be tested using BDNF antibody except for the HA tag. In Figure1E, a bigger scope shoud be shown to validate the pecificity of the virus expression.
2.GFAP was the astrocyte marker, what about the microglia?
3.Only single behavior, LD or TST was not adequate to draw the conclusion, more behaviors should be included, for example: SPT, FST, EPM, OFT, especially when your conclusion was different from the traditional perceive for mPFC BDNF was an beneficial factor in depression/anxiety.
4.How its effect on the maturation of neurons in these BDNFs treatment mice?

·

Basic reporting

Clear, unambiguous, professional English language used throughout.

R: Yes

Intro & background to show context. Literature well referenced & relevant.

R: Explore better the BDNF gene expression regarding it’s relation with anxiety-like behavior in the introduction section would be very valid. But in general, the current introduction is good, straight to the point and well structured.

Please find below some suggestions of references that would contribute to this intro section:

“Maternal stress during pregnancy, adverse early life experiences may predispose individuals to develop 56 psychopathology in adulthood Satinsky et al. (2021); Xia et al. (2023). Another factor that affects the 57 developing nervous system and is commonly encountered in perinatal medicine is antenatal glucocorticoid 58 therapy, which is prescribed to women at risk of preterm delivery. We have previously shown that 59 administration of the synthetic glucocorticoid dexamethasone causes neuronal apoptosis in the subiculum 60 of neonatal rat pups Lanshakov et al. (2016a).”

Weaver IC, Cervoni N, Champagne FA, et al. Epigenetic programming by maternal behavior. Nat Neurosci. 2004;7(8):847-854. doi:10.1038/nn1276

Szyf M, Weaver IC, Champagne FA, Diorio J, Meaney MJ. Maternal programming of steroid receptor expression and phenotype through DNA methylation in the rat. Front Neuroendocrinol. 2005;26(3-4):139-162. doi:10.1016/j.yfrne.2005.10.002

Meaney MJ, Szyf M. Environmental programming of stress responses through DNA methylation: life at the interface between a dynamic environment and a fixed genome. Dialogues Clin Neurosci. 2005;7(2):103-123. doi:10.31887/DCNS.2005.7.2/mmeaney

de Mendonça Filho EJ, Barth B, Bandeira DR, et al. Cognitive Development and Brain Gray Matter Susceptibility to Prenatal Adversities: Moderation by the Prefrontal Cortex Brain-Derived Neurotrophic Factor Gene Co-expression Network. Front Neurosci. 2021;15:744743. Published 2021 Nov 24. doi:10.3389/fnins.2021.744743

Discuss the environment importance through genome regulation/gene expression in the introduction would clarify the context surrounding BDNF expression, since it’s known specifically the exon IV may modulate PFC functions.

Please find below some paper suggestion in this matter:

Labonté B, Suderman M, Maussion G, et al. Genome-wide epigenetic regulation by early-life trauma. Arch Gen Psychiatry. 2012;69(7):722-731. doi:10.1001/archgenpsychiatry.2011.2287

Costa GA, de Gusmão Taveiros Silva NK, Marianno P, Chivers P, Bailey A, Camarini R. Environmental Enrichment Increased Bdnf Transcripts in the Prefrontal Cortex: Implications for an Epigenetically Controlled Mechanism. Neuroscience. 2023;526:277-289. doi:10.1016/j.neuroscience.2023.07.001

Sakata K, Woo NH, Martinowich K, Greene JS, Schloesser RJ, Shen L, Lu B (2009) Critical role of promoter IV driven BDNF transcription in GABAergic transmission and synaptic plasticity in the prefrontal cortex. Proc Natl Acad Sci USA 106 (14):5942–5947. https://doi.org/10.1073/pnas.0811431106.

Structure conforms to PeerJ standards, discipline norm, or improved for clarity.

R: Yes

Figures are relevant, high quality, well labelled & described.

R: Image is too small in Fig 1C. It is important to be more visible. I suggest making it bigger.

The WB membranes pics attached as supplement look good and original.

Raw data supplied (see PeerJ policy).

R: Yes. Well organized spreadsheets.

DNA data well supplied.

Experimental design

Methods described with sufficient detail & information to replicate:

R: In methods - behavior testing (line 213-231) is missing references for both tests, regarding the chosen parameters (lux used, analyzed variables, etc).

Validity of the findings

The anxiety-like behavior reported by the LDB and TST tests are valid contributions and benefit the literature in the context of BDNF and anxiety-like behaviors associated with its expression, since the majority of the previous studies used the Elevated Plus Maze (EPM) test for this purpose, considering its a more robust test for this kind of analysis.

Additional comments

The results described in the lines 301 to 303 are in accordance with the results in this study, but I suggest more caution in the conclusion (lines 478, 479) once this study miss more robust behavioral tests to assess the anxiety-like behavior, for example, a elevated plus maze test.

Reviewer 3 ·

Basic reporting

The manuscript is generally written in clear, professional language, though there are occasional instances where the phrasing could be improved for readability. For example, sentences such as "A small flexible 5W lamp was mounted above the light box, providing light to the center of the box" could be made more concise. The introduction is thorough, providing adequate background on BDNF, its role in brain development, and its implications for anxiety and depressive-like behaviors. However, the rationale for focusing specifically on the neonatal mPFC could be emphasized more clearly to justify the study's design. Additionally, a recent study on exosomes engineered to overexpress BDNF to treat depression could be introduced or discussed (Targeted Delivery of Engineered RVG-BDNF-Exosomes: A Novel Neurobiological Approach for Ameliorating Depression and Regulating Neurogenesis. Research,
4 Jul 2024, Vol 7,Article ID: 0402, DOI: 10.34133/research.0402)

Experimental design

The experimental design is rigorous, with detailed descriptions of constructs, animal handling, and statistical analyses. The use of RNA-seq to investigate transcriptomic changes adds depth to the study, and the behavioral assays are well-chosen to assess the effects of BDNF expression.

Validity of the findings

The methods are described in sufficient detail to allow for reproducibility, particularly with regard to the viral vector construction and behavioral assays. However, providing more detail on the RNA-seq processing and normalization steps would enhance the study's reproducibility.

Additional comments

The study's findings are intriguing and contribute valuable insights into the potential role of BDNF in psychiatric disorders. However, more discussion on the broader implications of these findings would be beneficial.

---

## Round 0.2 · accepted · Accept

The authors addressed all the comments raised by the reviewer, and the paper can now be accepted for publication.

·

Basic reporting

no comment

Experimental design

no comment

Validity of the findings

no comment

Additional comments

The authors have well addressed by concerns, and I have no further comments.